# The Effect of Integrating Service-Learning and Learning Portfolio Construction into the Curriculum of Gerontological Nursing

**DOI:** 10.3390/healthcare10040652

**Published:** 2022-03-31

**Authors:** Pei-Ti Hsu, Ya-Fang Ho, Jeu-Jung Chen

**Affiliations:** 1Department of Nursing, Ching Kuo Institute of Management and Health, Keelung 20301, Taiwan; 2School of Nursing, China Medical University, Taichung 406040, Taiwan; avonho97@mail.cmu.edu.tw; 3Department of Rehabilitation, Taiwan Adventist Hospital, Taipei 10556, Taiwan; lauraapplepie@gmail.com

**Keywords:** service-learning, learning portfolio, curriculum of gerontological nursing, attitudes toward aging, older people behavioral intention

## Abstract

Background: With the rapid increase in the aging population, a greater number of older individuals will require nursing care in the future. Therefore, it is important for nurses to be willing to engage in gerontological nursing. Nursing students must increase their experience in providing care to older people during their education and must receive education that improves their attitudes toward aging; this will help provide care to the older people, develop positive attitudes toward aging, and increase their empathy and willingness to provide care to older people after graduation. Hence, studies focused on improving the attitude of nursing students toward aging are urgently required. Methods: In this mixed-method experimental study, participants were interviewed individually and observed to better understand the connection between quantitative and qualitative data. Service learning and learning portfolio constructions were integrated in the gerontological nursing curriculum of an experimental group, whereas traditional gerontological nursing curriculum was provided to a control group. Quantitative data on the nursing students were collected using the attitudes toward aging scale (ATAS) and older people behavioral intention scale (OBIS) and analyzed using descriptive and inferential statistics. Result: From the pre- to the post-test, the average ATAS and OBIS scores of the experimental group increased significantly, reaching a statistically significant level. However, the results of the control group indicated that the educational intervention does affect the attitudes toward aging and older people behaviors. A qualitative analysis revealed that educational intervention can improve the students’ attitudes toward aging and older people behavioral intention. Conclusion: Our study results showed that integrating community older people service and learning portfolio construction into the curriculum can effectively improve the attitudes of nursing students toward aging and older people’s behaviors, thus providing substantial assistance to students intending to care for the older people in the future.

## 1. Introduction

With the rapid increase in aging population, more and more older individuals will need nursing care in the near future. Accordingly, nurses need to have the knowledge to provide expert care well as the intent to provide care to this group. During their education, nursing students should increase their experience in providing care to older people and should be educated about improving their attitudes towards aging; this approach will help them enhance their empathy, develop positive attitudes toward aging, and increase their willingness to provide better care to the older people after graduation [1,2,3,4]. Hence, studies that focus on improving the attitudes of nursing students toward aging are urgently required.

In a systematic review on nurses’ care for the older people, Rush et al. [5] reported that seven of the eight studies they reviewed indicate that nurses have a negative attitude against the older people that consequently influences the care provided. Most of the nursing students’ have negative views toward aging and assume, in general, that the older people are usually ill; have unclear verbal expression, mental fatigue, poor hearing, and poor mobility; are forgetful; and slow to respond [6,7]. However, studies have deduced that nursing students show more positive attitudes toward aging and have strong determination of serving the older people by accompanying them, helping them learn, and caring for them [6,8]. On the other hand, some studies have found that if students tend to have negative attitudes toward aging, providing students’ aging-related education and more opportunities of working with the older people will help them increase empathy, provide care to the older people, eliminate negative attitudes toward aging, be more willing to spend time with the older people, and increase their willingness and intentions of providing care and service to the older people [7,9].

Service learning is an experiential learning method designed with reciprocity and reflection as core factors. According to the service-learning method, service is as important as learning, thereby facilitating the completion of goals of both the service provider and the recipient [10]. Dinour and Kuscin [11] proposed that service learning can enhance students’ interpersonal development, practical application of coursework and classroom knowledge, reflection, critical thinking, and problem-solving skills. Additionally, it can foster student–community connections, support the community in solving problems, and nurture citizens continuously involved in community activities. Gardner and Emory [12] indicated that service learning enables students to appreciate various facets of individuals in the society, thereby helping them develop respect for others and toward different societal conceptions.

Learning portfolios are files used to record a student’s learning process and outcomes in a purposeful and organized manner. Projects in a learning portfolio are designed on the basis of teaching objectives and curriculum [13,14]. Learning portfolios can be outcome-based, such as project reports and process-based learning portfolios, such as activity notes, worksheets, reflection records, and instructor’s rating [13,15,16]. The use of learning portfolios in nursing education increases learners’ motivation [17], enables students to combine theory and practice [15,16,18], increases the availability of emotional support when nursing students encounter difficulties [16] and spaces for nursing students to express themselves [18], strengthens reflection skills of nursing students [16,19], and increases self-directed learning [19].

The objective of this study was to examine the effect of developing and integrating community service learning and learning portfolios into the curriculum of gerontological nursing on nursing students’ attitudes toward aging and behavioral intention toward the older people. Service learning and learning portfolios were the two teaching methods used to develop an alternative curriculum of gerontological nursing. Service learning enabled face-to-face interactions as well as discussions on topics such as aging and life experiences with the older people, and provided a better understanding of the living conditions of the older people to help students deepen their understanding of their life experiences, be aware of the issues faced by the older people, and develop empathy. During various stages of the curriculum, learning portfolios were used to enhance students’ motivation to learn and improve their critical thinking and reflection skills while inculcating positive attitudes toward aging, thereby improving their intention to provide care to the older people in the near future.

## 2. Materials and Methods

### 2.1. Study Design and Participants

This study employed a mixed-methods research design. The experimental study was conducted in which participants of experimental group were interviewed individually and observed to better understand the connections between quantitative and qualitative data. Figure 1 presents the study framework. Posters were posted on bulletin boards in the nursing department of a university in northern Taiwan, and students who volunteered to participate in the gerontological nursing curriculum for the first time were recruited. Next, students were randomized into experimental (*n* = 41) and control groups (*n* = 41). Service-learning and learning portfolio constructions were integrated in the gerontological nursing curriculum provided to the experimental group. The control group received courses delivered in a traditional method, which were conducted in the lecture methods with physiological aging, psychological aging, and social aging. The instructor for this research has more than 10 years of experience in gerontological nursing and severed as the instructor for both groups. This study has been approved by Taipei Hospital’s Institutional Review Board of the Ministry of Health and Welfare (TH-IRB-0019-0041). There was no risk to those who participated in this study. All participants who attended this program provided their informed written consent before the research commenced.

### 2.2. Curriculum of Experimental Group

As part of their curriculum, students in the experimental group had to visit community older people care stations and provide service to community-dwelling older people. The experience gained by providing service to them was to increase students’ understanding of the aging process. Learning portfolio construction (including aging concept map, student’s aging notes, aging experience worksheet, reflection log, and instructor’s reflection notes) was used to further enhance students’ understanding of the aging process and provide the older people with aids to resolve identified problems, which were modified, collected, and displayed. Finally, learning portfolios were completed, and students can review their learning process through the learning portfolio.

### 2.3. Procedure

The gerontological nursing curriculum provided to the experimental group initially included four visits to community older people care stations for service learning (once a week, 4 h per visit), followed by 10 weeks of learning portfolio integrated with gerontological nursing curriculum teaching (once a week, 2 lessons per session, with 100 min in total). The control group received courses delivered in a traditional method, which were conducted in the lecture methods with 5 weeks of physiological aging, 5 weeks of psychological aging, and 4 weeks of social aging (once a week, 2 lessons per session, with 100 min in total). Pretest (T0) data collection was conducted in two groups before the course commenced, and post-test (T1) data collection was conducted 1 week after the 14-week course had ended. In the experimental group, the curriculum was divided into five phases. Figure 2 presents the curriculum phases in detail.

Phase I: Community older people service learning

During community older people service learning, valuable information and data were gathered through in-depth observations, interviewing, and listening to the older people. This information and data could not have been gained by conventional face-to-face interviews alone. In weeks 1, 2, 3, and 4, the courses focused on community older people service learning. During the course activities, students conducted in-depth observations and interviewed the older people to understand their physiological, psychological, and social aging processes and life experiences, and concept maps for aging were developed after interviews.

Phase II: Output phase

The stage was the output stage in which portfolio objectives, implementation tasks, and assessment criteria were used to confirm the details of the implementation items and the type and content of the projects collected. On week 5 of the course, aging-related education was provided. In weeks 6, 7, and 8 of the course, learning portfolios for physiological, psychological, and social aging processes and life experiences of the older people were developed.

Phase III: Reflection phase

In week 9, students reviewed whether their work had met the portfolio assessment criteria. Moreover, reflection and recording were performed. The instructor determined and recorded whether students’ work had met the criteria and provided feedback.

Phase IV: Services and celebration: Creation of aids for the older people

In weeks 10, 11, 12, and 13, students examined some of the potential issues faced by the older people in terms of physiological, psychological, and social aging processes using learning portfolios and created aids to help solve aging-related problems. This enabled students to provide services and improve their behavioral intentions toward the older people.

Phase V: Presentation phase

In week 14, publication, exhibition, and sharing of learning portfolio outcomes and work was performed, and post-test data was collected within 1 week after the end of the course.

### 2.4. Learning Portfolio

The learning portfolio tool used in this study included concept maps for physiological aging, psychological aging, and social aging; student’s aging notes; aging portfolio worksheet; reflection log; and instructor’s reflection notes. The learning portfolios designed concept maps for physiological, psychological, and social aging; student’s aging notes; and aging portfolio worksheet mainly based on the module objectives. The services provided to the community older people and courses were used to design reflection logs and instructor’s reflection notes. Two experienced gerontological nursing experts were invited to provide comments for revising the contents and expression of the preliminarily drafted learning portfolios; the pilot learning portfolio was created after making suitable modifications. A total of 34 students were included in the pilot testing and completed all the formal learning portfolios. Table 1 shows the learning portfolios used in different weeks.

#### 2.4.1. Concept Maps for Physiological, Psychological, and Social Aging

During weeks 1–4 of the community older people service-learning module, students were asked to create concept maps to understand the conceptual framework and alternative conceptions. After each activity, the concept maps were evaluated, and three concept maps were drawn in total. The objective was to understand the overall concept and change of concept of students through the concept maps.

#### 2.4.2. Student’s Aging Notes

Students would express their learning experiences, knowledge, and insights they gained from every teaching module through handwritten notes, which, in turn, would help develop good habits of thinking and summarizing and would motivate them to learn more effectively.

#### 2.4.3. Aging Portfolio Worksheet

The investigator drafted worksheets based on the course content so that students could share their experiences with each other, and through the observation, inquiry, recording, and other processes required in the worksheets, students could personally obtain knowledge and experiences. All teaching modules were compiled into a complete learning portfolio worksheet; in total, there were five portfolio worksheets, namely, physiological aging worksheet, psychological aging worksheet, social aging worksheet, reflection worksheet, and aging-related problem worksheet.

#### 2.4.4. Reflection Log

The reflection log mainly focused on deepening students’ understanding of their learning process, gaining a clear understanding of their strengths and weaknesses as learners and the most effective learning strategy for them, developing the ability to solve learning-related problems, understanding the importance of self-assessment in the process of self-growth, and appreciating their own strengths.

#### 2.4.5. Instructor’s Reflection Notes

The instructor’s reflection notes allowed the instructor to focus on their own teaching activities and conduct serious self-review, feedback, control, regulation, and analysis of the generated outcomes as well as to examine their teaching philosophy and behavior and decision-making ability.

### 2.5. Measures

A mixed-methods research allows researchers to examine research issues more comprehensively, observe phenomena from different research perspectives, and discover the complex relationships between different aspects of research issues [20]. Thus, this study adopts a mixed method to collect data and analyzes intervention effectiveness with a structured questionnaire along with an interview. The qualitative interview is mainly to understand the process of students’ impression changes toward aging. The effectiveness of the intervention was assessed using a structured quantitative questionnaire prepared in accordance to the relevant literature as well as the study’s framework. The questionnaire consisted of three major parts—personal characteristics, attitudes toward aging scale (ATAS) scores, and older people behavioral intention scale (OBIS) scores. The questionnaire was subjected to professional evaluation, pilot testing, and revision to obtain an official questionnaire. The qualitative effectiveness scale was used for conducting qualitative interviews and observations; the scale helped comprehend the learning process of nursing students through the learning portfolio. The scales used in the study are described in the following sections.

#### 2.5.1. Personal Characteristics

The personal characteristics include age, sex, place of residence, religion, experience contact with the older people. The personal characteristics was the control variables for this study. Experience contact with older people contains nine items that are scored on a scale of 1 to 5, with the lowest and highest possible total scores being 9 and 45, respectively. A higher score indicates that the participant has higher degree of contact experience with older people. The scale has a Cronbach’s α value of 0.85.

#### 2.5.2. ATAS

Attitudes toward aging was the effectiveness variable of this study and was measured by ATAS. The ATAS used in this study was a researcher-developed structured questionnaire. The preliminary structured questionnaire was drafted according to the study’s framework and related literature [21,22,23]. The questionnaire underwent expert review to determine its validity, pilot testing, and revision to obtain an official questionnaire. The validity of the content of the ATAS was examined by three nursing experts and one health education expert. The CVI score of the ATAS is 0.87. The nursing students’ ATAS contains 19 questions, and the scale is divided into 3 dimensions: aging-related awareness, feelings toward older people, and interpersonal interactions with older people. A 5-point Likert scale was used, and scores ranged from 1 (highly disagree) to 5 (highly agree). The higher the score, the more positive the students’ attitude toward older people. The Cronbach’s α values of the overall scale and the subscales are 0.91, 0.90, 0.89, and 0.91, and the intraclass correlation coefficient [ICC] is 0.84, indicating that this questionnaire has great reliability and validity.

#### 2.5.3. OBIS

Older people behavioral intention was the effectiveness variable of this study and was measured by OBIS. The OBIS was also a researcher developed structured questionnaire. The preliminary structured questionnaire was drafted according to the study’s framework and related literature [22,23]. The questionnaire underwent expert review to determine its validity, pilot testing, and revision to obtain an official questionnaire. The validity of the content of the OBIS was examined by three nursing experts and one health education expert. The CVI score of the ATAS is 0.89. The scale contains 15 questions in which nursing students’ older people behavioral intention was divided into four dimensions, namely companionship, services, learning, and employment. A 5-point Likert scale was used, and scores ranged from 1 (highly disagree) to 5 (highly agree). The higher the score, the higher the students’ older people behavioral intention. The Cronbach’s α values of the overall scale and the subscales are 0.92, 0.91, 0.92, 0.91, and 0.92, and the intraclass correlation coefficient [ICC] is 0.93, indicating that this questionnaire has great reliability and validity.

#### 2.5.4. Qualitative Data Collection

Participation and observation details, individual interviews, and students’ reflection logs were used to collect qualitative data. The participation and observation details included students’ class participation status and group discussion details. Two individual interviews were conducted in the instructor’s office after the pretest and post-test questionnaires were completed. The interviews lasted for 30 min and were conducted by the same study staff to ensure consistency. The participants provided their consent for recording the interview, and a transcript was obtained within 24 h. The control group in this study was used to examine the effect on the outcomes of interest in quantitative research but not in qualitative research. Therefore, interviews were not arranged in the students of the control group.

### 2.6. Data Processing and Analysis

IBM SPSS software ver. 23.0 (IBM Corp., Armonk, NY, USA) was used to perform statistical analyses. All data were presented as mean and standard deviation or frequency and percentage. Chi-square test and *t*-test were used to compare whether categorical and continuous variables differed between groups. Paired *t*-test was used to evaluate intragroup differences between pretest and post-test mean scores. The students’ pretest results were used as covariates and groups were used as independent variables for performing one-way analysis of covariance to compare post-test differences between the experimental and control groups. In this study, *p* < 0.05 indicated statistical significance.

Thematic analysis was used for qualitative data. The source data was sorted and coded by two study staff. The study staff first read the transcript several times to develop initial open codes before the concepts were compared and grouped into themes. Finally, common subthemes were summarized to form themes. Two study staff conducted independent reviews and periodically held meetings for discussion, thereby strengthening the stringency of the process.

## 3. Results

### 3.1. Personal and Baseline Characteristics of Participants

A total of 82 students participated in all stages of this study. There are 41 students in the experimental group, and 41 students in the control group. The average age of the subjects in the experimental group is 20.04 (±0.20), and the average age of those in the control group is 20.05 (±0.22). Before the intervention, there were no significant differences in the scores of the two groups in personal characteristics, ATAS, and OBIS (Table 2).

### 3.2. Pretest and Post-Test Changes in the Experimental Group after Educational Intervention

From Table 3, it can be deduced that the post-test (T1) mean scores for total ATAS score, aging-related awareness, feelings toward the older people, interpersonal interactions with the older people, and total OBIS score of the experimental group were higher than the pretest (T0) scores. The paired *t*-test revealed that all post-test scores of the experimental group were higher than the pretest scores, and the difference was statistically significant (*p* ≤ 0.001).

### 3.3. Changes in Post-Tests of the Experimental Group and the Control Group after the Education Intervention

Before the covariate variable analysis, the intragroup homogeneity test of the regression coefficients was performed. The results showed that the regression coefficients within each group are homogeneous (*p* > 0.05), which is in line with the assumption of the covariate variable analysis. All pretest results (age, gender, socioeconomic status, place of residence, religious beliefs, frequency of interaction with older people, attitudes toward aging, and older people behavioral intention) were used as covariates to understand the effectiveness of educational intervention in improving the students’ attitudes toward aging and older people behavioral intention. From Table 4 and Table 5, it can be seen that there were significant differences in intervention effects for “aging-related awareness” after educational intervention (F = 18.36, *p* < 0.001), and the experimental group had a higher score (28.87 ± 0.15) than the control group (27.93 ± 0.15). There were significant differences in teaching effects for “feelings toward older people” after educational intervention (F = 50.03, *p* < 0.001), and the experimental group had a higher score (25.45 ± 0.25) than the control group (22.93 ± 0.25). There were significant differences in teaching effects for “interpersonal interactions with older people” after education intervention (F = 47.27, *p* < 0.001), and the experimental group had a higher score (23.98 ± 0.32) than the control group (20.88 ± 0.32). There were significant differences in “total ATAS score” after education intervention (F = 58.20, *p* < 0.001), and the experimental group had a higher score (78.29 ± 0.60) than the control group (71.78 ± 0.60). There were significant differences in teaching effects for OBIS after education intervention (F = 28.62, *p* < 0.001), and the experimental group had a higher score (65.57 ± 0.67) than the control group (60.45 ± 0.68). These results showed that the educational intervention has teaching effects in “attitudes toward aging” and “older people behavioral intention”.

### 3.4. Witnessing One’s Learning Growth Process through Educational Intervention

Individual interviews of students in the experimental group were conducted to evaluate the learning processes of study participants, including activity content, participation process, and reflection, which were used for learning process analysis and described in four processes.

Process 1—Most students had negative attitudes toward aging before the intervention

Before the intervention, the students’ attitudes toward aging were slow movement, lack of vitality, frail, and sickly. A student described, “*In my memory, all the older people were skinny, physically unwell, and would walk with a hunchback, and I felt that they walk in an unsteady manner that causes them to fall*…” (Line 250–253). Generally, most students believed that the older people were often have a patronizing attitude and are difficult to get along. A student mentioned that “*The older people that I usually meet in the city have left a bad impression, e.g., cutting queues for buses, insisting on squeezing in and snatching seats, and tending to argue with others*…” (Line 650–652). Another student mentioned that “They will start to nag constantly when they see things that they do not agree with or do not like.” (Line 156–157). Many students mentioned that the older people are frequently disconnected from the society. A student mentioned that “*Some retirees like to stay at home as they do not like to participate in activities. The older people also do not participate in activities due to decreased productivity and poor work efficiency*”. (Line 167–169).

Process 2—Community older people service-learning process

After students in the experimental group participated in community older people service activities, they found that not all the older people were “frail and sick”. A student mentioned that “*Through the community older people service learning, I observed that although the older people’s physical functions gradually degenerate with age, the aging rate differs from person to person. Many older people still have good spirits, are mentally active, and can walk fast. Therefore, age should not be used to determine an older people individual’s function, physical strength, and mental status*”. (Line 920–924). In addition, there was no “generation gap and difficulty in getting along” as previously assumed. A student mentioned that “*I was nervous when I had to go for community service learning in week 1. After I met the community-dwelling older people, I felt that they were easy to get along with and well-mannered, which is totally different from my impression of the older people, i.e., they would feel that they are more experienced due to their old age and are difficult to get along with, but my original impression has changed*”. (Line 1050–1055). The older people show concern for young people. A student mentioned that “*Although some the older people often tell you that you cannot do certain things, they are actually concerned about young people*”. (Line 1137–1139). After service learning, the relationship between students and the older people improved, and there was a sense of mutual understanding. Students viewed the older people as friends or partners. A student mentioned that “*They were never stingy and were willing to share their knowledge and experiences. When facing these selfless people, I gradually opened my heart to trust others. In the last few service-learning sessions, I truly felt that they are friends and partners and not mere elders*” (Line 1336–1340). The students provided feedbacks wherein they stated that they understood how to communicate with the older people through interaction, which, in turn, has increased their intention to serve the older people in the near future. A student mentioned that “*In this service learning, I learned to get along with the older people, which enabled me to learn techniques for communicating with the older people. I feel empathy for them, and I like to meet the older people. In the future, I hope to work in the older people care*”. (Line 1515–1518).

Process 3—Learning portfolio construction and reflection process

During the portfolio construction and reflection process, students realized that their previous impression of the older people was incorrect. They believed that although many older people are above 65 years of age, they still possess a good working competency. Additionally, the retired older people above 65 years old can work in a new job after retirement. A student mentioned that “*Many retired older people are still volunteering and show good work performance*”. (Line 1756–1757); another student mentioned that “*Many older people arranged volunteering activities for themselves, and the older people have rich life experiences that they can share with young people*”. (Line 1772–1774). Students reflected that not all old people are the same. A student mentioned that “*The older people are individuals like us, and they previously had lives similar to ours*”. (Line 1876–1877). “*Although the physical functions of the older people gradually decrease with age, the rate of aging differs from people to people, and I have positive attitudes toward aging processes and changes*”. (Line 1925–1928). Students also thought that being old does not mean being ill, and a student mentioned that “*Being old does not necessarily mean that one is ill. Many older people are still physically fit and active*…” (Line 2014–2016); another student mentioned that “*Nowadays, being old does not mean that one is unhealthy, and many older people live longer and are healthier than the older people in the past*”. (Line 2213–2215). Students also reflected that influencing factors of psychological adjustment ability are an individual’s attitudes, motivation, and health status. Although a deterioration in adjustment capacity may occur in old age, it is frequently linked with physiological imbalance, depression, loss of social support, and environmental factors. A student mentioned that “*I felt that most of the older people that I talked to are cheerful and easy to talk to. They are psychologically active, lives happy lives every day, and are exemplars that we can learn from*”. (Line 2366–2368). Students stated that through portfolio content construction and reflection process, they have developed more positive attitudes toward aging and empathy and are willing to provide services to the older people in the near future. A student mentioned that “*This curriculum enabled me to better understand the older people and feel more empathy for them, which can help me to know how to get along with the older people in my future job, and I would like to work in the older people care*”. (Line 2451–2454). Another student mentioned that “*This curriculum enabled me to think from the perspective of the older people…I will not be afraid of working with the older people if I get an opportunity to work in occupations that contact and provide services to the older people*”. (Line 2525–2528). With regard to learning, students felt that learning portfolio construction improves their motivation to learn, and they have become more proactive in terms of learning. They can actively think of solutions to problems, better apply theory in practice, and continuously reflect while learning. A student mentioned that “*Learning portfolio homework enabled me to have greater motivation and learn proactively. I am better at applying theory in clinical practice, and I continuously reflect on the learning process and experience*”. (Line 2555–2558).

Process 4—Older people assistive device creation activity

After students in the experimental group underwent the older people assistive device creation activity, they enjoyed helping the older people and were more willing to care for the older people. A student mentioned that “*I felt that after observing discomfort in the older people during my curriculum and designing aids for the older people, I felt a sense of achievement that I can help the older people improve their quality of life, and I deeply felt the joy of helping people*”. (Line 2625–2628). Another student mentioned that “*After creating the assistant device for the older people, it makes me more willing to take care of the older people*”. (Line 2713–2715).

### 3.5. Instructor’s Teaching Reflection

The integration of service learning into curriculum and use of interactive teaching methods enabled the instructor to observe that students had become more interested in learning and opportunities for interacting with the older people are expected to increase. The knowledge and understanding of the aging process will help increase students’ readiness for working in a rapidly aging society and their positive attitudes toward the aging society. Creating learning portfolios during the curriculum enabled students to record the content, development process, and outcomes of learning, as well as observe and assess their learning progress and reflect on their results while learning.

## 4. Discussion

In this study, service learning and learning portfolio construction were the two methods used to improve nursing students’ attitudes toward aging and older people behaviors. The results revealed that the integration of the two methods into the gerontological nursing curriculum successfully improved the nursing students’ positive attitudes toward aging and older people behaviors.

### 4.1. Service-Learning

This study started with the service-learning method, in which students observed and conversed with the older people attending community older people services. This effectively resulted in positive changes in students’ attitudes toward the older people, aging-related awareness, feelings toward the older people, and interpersonal interactions with the older people as well as improved the older people behavioral intention, which included accompanying the older people, servicing the older people, helping the older people to learn, and providing the older people care in the near future. These results were similar to the study of Hou [24] who integrated service learning into the curriculum and determined that aging-related knowledge, attitudes toward getting along with the older people, and intention of providing services to the older people in the near future increased among university students who participated in the provision of services to the older people. The qualitative analysis of this study supported these findings, thereby revealing that the integration of service learning into gerontological nursing curriculum can improve nursing students’ attitudes toward the older people. Nursing students who received education of service learning had a positive response to the course. The students mentioned that service learning can help them get along with the older people, convert their views of the older people into positive attitudes toward aging, and view the older people not just as their elders but partners.

This study emphasized the equal importance of servicing and learning objectives and integrated servicing into professional curriculum. Besides professional skills, the curriculum content also included increase the opportunity to provide services to older people. By caring for community-dwelling older people, performing needs assessment, and formulating a service, execution, and assessment plan, students learned to assist the community-dwelling older people with their needs. This ensured that both goals of teaching and promoting healthy aging among the community-dwelling older people are met.

### 4.2. Learning Portfolio Construction

The study results revealed that this curriculum employed a learning portfolio construction. The qualitative analysis revealed that learning portfolio construction can increase students’ motivation to learn. They would be more proactive in terms of learning, can actively think of solutions to problems, can effectively apply theoretical knowledge in practice, continuously reflect while learning, and provide older people care in the near future. Kathleen et al. [17] employed learning portfolio to increase nursing students’ motivation to learn and linked them with future employment opportunities. Margery et al. [25] determined that learning portfolios can deepen medical students’ understanding of learning outcomes and students became more proactive in terms of learning. Gordon [26] integrated learning portfolios into curriculums for medical students and deduced that such portfolios are beneficial for thinking and reflecting clinical experience. Kariman and Moafi [27] used learning portfolios for midwifery students, and students felt that the incorporation of such portfolios increased their participation in the learning process and helped them apply theoretical knowledge in practice. Jenkins [13] and Tahriri et al. [14] highlighted that learning portfolios demonstrated the learning process, progress, and outcomes in a certain field and allowed students to continuously reflect while learning.

The study results revealed that students’ attitudes toward aging improved, in addition to their aging-related awareness, feelings toward the older people, interpersonal interactions with the older people, and older people behavioral intention, which includes accompanying the older people, providing service to them, helping them learn, and providing older people care in the near future. These results were similar to the study conducted by Assadi et al. [18], in which learning portfolios were integrated into nursing education, and it was determined that such portfolios effectively improved the knowledge, attitudes, and skills of nursing students.

The two methods of integrating service-learning and learning portfolio construction into the curriculum of gerontological nursing may require more time and effort for teachers. Through this research, we found that teachers need to learn about teacher-related teaching methods before the course, design the course content during the summer break, and then discuss with some other gerontological nursing teachers to avoid an obstacle of the course. We believe that relevant teaching assisting resources are also necessary, such as teaching assistants, etc., so that these two teaching methods can be promoted immediately. Otherwise, even if there is a good method, it may consume more time and energy for teachers, this will make most of the teachers to hold back.

### 4.3. Limitations

Although this study reported important results of integrating service-learning and learning portfolio construction into gerontological nursing, the results of improved attitudes toward aging and older people behavioral intention are limited because nursing students from only one school were included. Therefore, further research is needed to determine whether these findings are consistent and transferable to other student populations when applied to other related healthcare fields. There was no follow-up period in this study, and we were unable to investigate the long-term effectiveness of the intervention. Therefore, in-depth investigation must be carried out in future studies.

## 5. Conclusions

This study examined the effects of integrating service learning and learning portfolio construction into gerontological nursing curriculum on nursing students’ attitudes toward aging and older people behavioral intention. The study results showed that the integration of community older people service learning and learning portfolio construction into curriculum can effectively improve nursing students’ attitudes toward aging and older people behavioral intention and therefore provides substantial assistance to students having intentions to participate in the clinical care for older people in the future. Therefore, service learning and learning portfolios can be used as bases to design more in-depth teaching content. We recommend that learning portfolios and service-learning lesson plans be integrated in future gerontological nursing curriculum to better meet nursing education needs. Future study can apply integrated service learning and learning portfolio construction to students of different related medical fields, and can increase the follow up time of the research to understand the long-term result.

## Figures and Tables

**Figure 1 healthcare-10-00652-f001:**
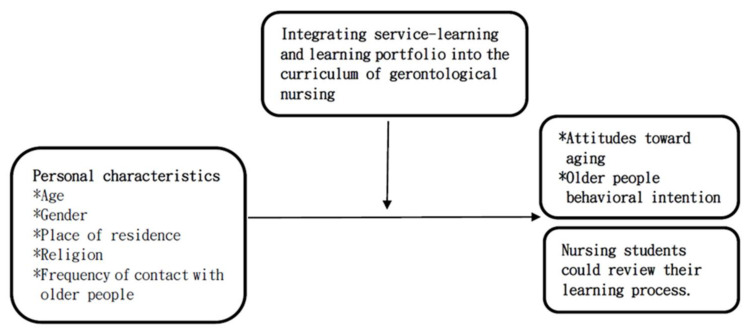
Study Framework.

**Figure 2 healthcare-10-00652-f002:**
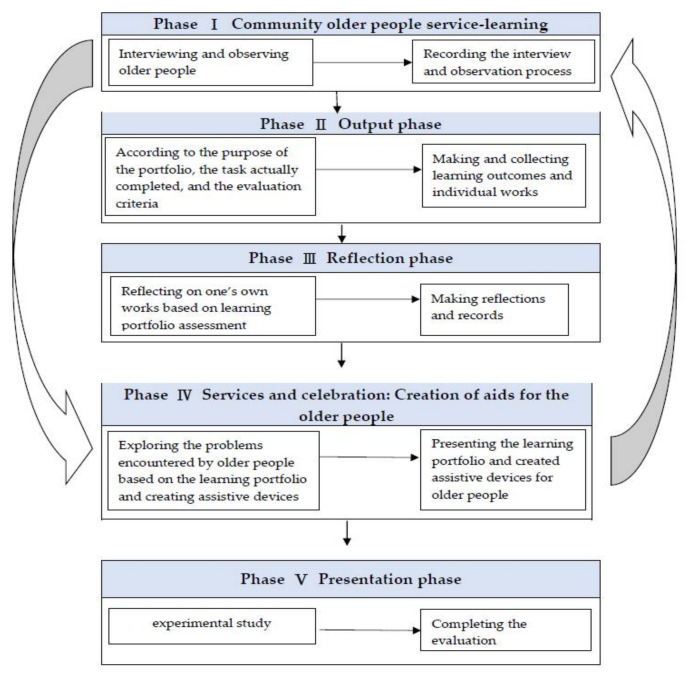
Curriculum stages for the experimental group.

**Table 1 healthcare-10-00652-t001:** Curriculum content.

Week	Experimental Group			Control Group
	Phase	Unit	Teaching and Learning Activities	Traditional Courses
1	I	Service learning with the community older people (I)	**Discussion**: Observing and interviewing the community-dwelling older people during service learning and discussing attitudes toward older people. **Learning portfolio**: Learning notes **Analysis**: Recording and analyzing the interview and observation process	**Method**: Lecture **Content**: Physiological aging
2		Service learning with the community older people (II)	**Discussion**: Observing and interviewing the community-dwelling older people during service learning and discussing physiological aging in older people. **Learning portfolio**: Physiological aging concept maps **Analysis**: Recording and analyzing the interview and observation process	**Method**: Lecture **Content**: Physiological aging
3		Service learning with the community older people (III)	**Discussion**: Interviewing the community dwelling older people and discussing psychological aging in older people. **Learning portfolio**: Psychological aging concept maps **Analysis**: Recording and analyzing the interview and observation process.	**Method**: Lecture **Content**: Physiological aging
4		Service learning with the community older people (IV)	**Discussion**: Interviewing the community dwelling older people and discussing social aging in older people. **Learning portfolio**: Social aging concept maps **Analysis**: Recording and analyzing the interview and observation process.	**Method**: Lecture **Content**: Physiological aging
5	II	Aging course activities	**Discussion**: Discussing aging in older people. **Learning portfolio**: Aging notes, aging portfolio worksheets, and teacher’s reflection notes **Analysis**: Recording and analyzing the learning portfolio, self-evaluation, and peer review and feedback.	**Method**: Lecture **Content**: Physiological aging
6		Establishment of a physiological aging portfolio	**Discussion**: Discussing physiological aging in older people. **Learning portfolio**: Physiological aging notes, worksheets, and teacher’s reflection notes **Analysis**: Recording and analyzing the learning portfolio, self-evaluation, and peer review and feedback.	**Method**: Lecture **Content**: Psychological aging
7		Establishment of a psychological aging portfolio	**Discussion**: Discussing psychological aging in older people. **Learning portfolio**: Psychological aging notes, worksheets, and teacher’s reflection notes **Analysis**: Recording and analyzing the learning portfolio, self-evaluation, and peer review and feedback.	**Method**: Lecture **Content**: Psychological aging **Midterm exam**: Written test
8		Establishment of a social aging portfolio	**Discussion**: Discussing social aging and life experience of older people. **Learning portfolio**: Social aging notes, worksheets, and teacher’s reflection notes **Analysis**: Recording and analyzing the learning portfolio, self-evaluation, and peer review and feedback.	**Method**: Lecture **Content**: Psychological aging
9	III	Reflection on the learning portfolio	**Written reflection**: Students assess whether their works meet the standard, based on the evaluation criteria for the portfolio, and reflect on and record the result. Students discuss their thoughts on the process of establishing a learning portfolio and what this experience means to them. The teacher assesses and records whether students’ works meet the standard and provides feedback accordingly. **Learning portfolio**: Reflection log	**Method**: Lecture **Content**: Psychological aging
10	IV	Helping older people find solutions	Discussing the problems of older people encounter during the process of aging. Discussing direction and incipient ideas for the creation and design of age-friendly products. **Learning portfolio**: Aging problem worksheets **Analysis**: Recording and analyzing the learning portfolio.	**Method**: Lecture **Content**: Psychological aging
11–12	V	Helping older people solve problems	Designing and making age-friendly products. **Learning portfolio**: Aging notes **Analysis**: Recording and analyzing the learning portfolio, self-evaluation, and peer review and feedback.	**Method**: Lecture **Content**: Social aging
13		Accomplishment of the task	Sharing age-friendly products created by each group and giving comments. Completing modifications to assistive devices for older people. **Learning portfolio**: Reflection log and teacher’s reflection notes	**Method**: Lecture **Content**: Social aging
14		Presentation and celebration	Presenting, exhibiting, and sharing the outcomes of learning portfolios and created works and collecting data on the post-test within one week of the course ending.	**Method**: Lecture **Content**: Social aging **Final exam**: Written test

**Table 2 healthcare-10-00652-t002:** Personal characteristics and baseline data of nursing students by group (*n* = 82).

Variables	Experimental Group (*n* = 41)		Control Group (*n* = 41)		*x*^2^/*t*	*p*
	*n* (%)	Mean (SD)	*n* (%)	Mean (SD)		
Age		20.04(0.20)		20.05 (0.22)	0.02	0.98
Gender					0.12	0.72
Male	15 (36.6)		14 (34.1)			
Female	26 (63.4)		27 (65.9)			
Place of residence					0.79	0.37
Keelung	21 (51.2)		25 (61.0)			
Non-Keelung	20 (48.8)		16 (39.0)			
Religion					2.48	0.11
Yes	13 (31.7)		20 (48.8)			
No	28 (68.3)		21 (51.2)			
Experience contact with older people		26.68 (9.40)		26.56 (7.60)	0.06	094
ATAS total		72.60 (7.54)		70.51 (8.13)	1.21	0.23
* Aging-related awareness		28.00 (2.60)		27.75 (3.19)	0.37	0.71
* Feelings toward older people		23.39 (3.33)		22.56 (3.44)	1.11	0.27
* Interpersonal Interactions with older people		21.21 (4.31)		20.19 (3.85)	1.13	0.26
OBIS		56.41 (7.85)		53.46 (9.46)	1.53	0.12

ATAS—Attitudes Toward Ageing Scale; OBIS—older people behavior intention Scale (OBIS).

**Table 3 healthcare-10-00652-t003:** Efficacy variable score paired *t*-value test after educational intervention.

Group	Variables	Pretest (T0)	Post-Test (T1)	*t*
		Mean (SD)	Mean (SD)	
Experimental	**ATAS total**	72.60 (7.52)	79.07 (6.50)	7.51 ***
	* Aging-related awareness	28.00 (2.61)	28.97 (2.41)	5.76 ***
	* Feelings toward older people	23.39 (3.30)	25.73 (2.42)	6.49 ***
	* Interpersonal Interactions with older people	21.21 (4.31)	24.36 (3.52)	6.54 ***
	**EBIS**	56.41 (7.82)	66.60 (8.10)	11.04 ***
Control	**ATAS total**	70.51 (8.13)	71.78 (7.54)	0.71
	* Aging-related awareness	27.75 (3.19)	27.95 (2.60)	0.30
	* Feelings toward older people	22.56 (3.44)	23.27 (3.33)	0.96
	* Interpersonal Interactions with older people	20.19 (3.85)	20.81 (4.31)	0.62
	**EBIS**	53.46 (9.46)	53.46 (9.46)	1.81

*** *p* < 0.001.

**Table 4 healthcare-10-00652-t004:** One-way ANCOVA analysis.

Variables	Variance	*SS*	*df*	*MS*	*F*
**ATAS total**	Between groups	851.92	1	851.92	58.20 ***
	Within groups	1156.36	79	14.63	
	Total	466,945.000	82		
* Aging-related awareness	Between groups	18.07	1	18.07	18.36 ***
	Within groups	77.75	79	0.98	
	Total	66,741.00	82		
* Feelings toward older people	Between groups	128.02	1	128.02	50.03 ***
	Within groups	202.11	79	2.55	
	Total	48,812.00	82		
* Interpersonal Interactions with older people	Between groups	191.96	1	191.96	47.27 ***
	Within groups	320.78	79	4.06	
	Total	42,668.00	82		
**OBIS**	Between groups	521.81	1	521.81	28.62 ***
	Within groups	1439.90	79	18.227	
	Total	331,081.00	82		

*** *p* < 0.001.

**Table 5 healthcare-10-00652-t005:** Adjusted post-test score averages of the two groups.

Group	Pretest (T0)	Before Adjusted Post-Test (T1)	After Adjusted Post-Test (T1)
	Mean(SD)	Mean (SD)	Mean (SE)
**Experimental**			
**ATAS total**	72.60 (7.54)	79.07 (6.59)	78.29 (0.60)
* Aging-related awareness	28.00 (2.60)	28.98 (2.50)	28.87 (0.15)
* Feelings toward older people	23.39 (3.33)	25.73 (2.43)	25.45 (0.25)
* Interpersonal interactions with older people	21.21 (4.31)	24.36 (3.80)	23.98 (0.32)
**OBIS**	56.41 (7.85)	66.61 (8.10)	65.57 (0.67)
**Control**			
**ATAS total**	70.51 (8.13)	71.00 (6.36)	71.78 (0.60)
* Aging-related awareness	27.75 (3.19)	27.83 (2.8)	27.93 (0.15)
* Feelings toward older people	22.56 (3.44)	22.65 (3.07)	22.93 (0.25)
* Interpersonal interactions with older people	20.19 (3.85)	20.51 (3.80)	20.88 (0.32)
**OBIS**	53.46 (9.46)	59.41 (6.7)	60.45 (0.68)

## Data Availability

The data presented in this study are available on request from the corresponding author.

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
