# Peer review of "The Effect of Integrating Service-Learning and Learning Portfolio Construction into the Curriculum of Gerontological Nursing"

_healthcare, 2022, doi:10.3390/healthcare10040652_

Round 1

Reviewer 1 Report

This paper studies how to improve the willingness of nursing graduates to engage in nursing care for the elderly. This topic has a certain degree of significance for improving the nursing care of the elderly, and has a certain academic research value. However, some defects or problems in the revised paper still need to be improved. The details are as follows:

  1. From the overall research and experimental process of this article, there are basically not too many problems, but there is redundancy in the description of some languages in the article, and it is necessary to be more concise in the language description. For example, in Section 3.4, the description of the middle school students is similar to " "My impression is that the elderly often sit and watch TV to sleep after retirement, and their energy is not good. "The language is somewhat redundant.
  2. Most of the experimental subjects selected in this paper are students from the nursing department of a college in northern Taiwan. Except for the defects mentioned in Section 4.3 in the text, the experimental subjects also have obvious regional characteristics. A description of the impact of the experimental results.
  3. There are small errors in the typesetting in this article. For example, in 2.4.5 after 2.4.4, there is a small error in the serial number mark, which needs to be corrected.
  4. The relationship between OBIS, ATAS, personal characteristics and experimental results in the questionnaire components used in the experiment of this paper needs to be explained.
  5. Please add future research directions in the conclusion of this paper.
  6. There is insufficient research in this paper on the role of study profiles in improving nursing graduates' willingness to engage in elder care.

Reviewer 2 Report

Integrating Service-Learning into the Curriculum of Gerontological Nursing: the Construction of a Learning Portfolio and Reflection

Reviewer comments:

Overall

Manuscript title: The title is confusing.  The authors indicate that the intervention was service learning and portfolio development.  So unclear why the “construction of a Learning Portfolio and Reflection” follow the colon.

Editorial: The word usage throughout the manuscript needs to be improved.  A thorough review of the sentence structure is needed.

Control group: More details on the control group education is needed. What was content of the course as well as the learning activities? Was the instructor the same for both groups? Rationale for not conducting interviews of the control group members is needed.

Repetition of the intervention information is not necessary. Be more concise.

Measures: Why did the authors create questionnaires specific for the study?  Why were standard measures not used? 

Until more information on the control group education is provided the conclusions of the study as presented by the authors is not meaningful – the intervention used resulted in higher scores – as compared to what we do not know. 

More specific comments follow.

Examples of the editorial issues in the manuscript follow (note these are examples and do not encompass all of the editorial issues of the manuscript)

Page 4 line 54

Most junior college students have negative views toward aging and generally believe that  older people are ill; have unclear verbal expression, mental fatigue, poor hearing, and  poor mobility; are forgetful; and respond slowly [3,6].

The above statement refers to “junior college students”.  Neither of the cited references have junior college students as subjects.

Page 4 Line 57

Studies have found that junior college students with more positive attitudes toward aging have stronger intentions of serving older people by accompanying them, helping them learn, and caring for them [7– 9].

Two of the 3 references cites for the above statement do have junior college students as subjects.  It is also noted that reference 6 and 9 are the same reference.

Page 8 Line 234

Thus, this study  adopts a mixed-method to collect data and analyzes intervention effectiveness with a structured questionnaire.

Authors need to add to this sentence: along with an interview.

This paragraph continues: The qualitative interview is mainly to understand the process of students’ impression changes toward This enabled them to provide services to older people and improve their behavioral intention toward older people.. after the education intervention. The effectiveness of the intervention was assessed using a structured quantitative questionnaire prepared in accordance to the relevant literatures as

There are incomplete sentences and thoughts in the above paragraph.

Page 8 Line 250

Another incomplete sentence: The personal characteristics include age、sex、place of residence、Religion、 experience contact with the older people.

Examples of the method questions of the study follow:

Page 4 line 134

Traditional teaching methods were employed for the control group.

“Traditional teaching methods” needs to be described. 

General comment on this this.  A lot of repetitive information is provided on the experimental teaching.  Need to describe the control group education.

Page 8

Was the instructor the same person for both the control and experimental courses?

Only the experimental group participants were interviewed.  Why were the control group participants not interviewed? 

Page 9

The authors developed their own questionnaires for this study.  Internal reliability is provided and they note “expert” review.  There is no mention of validity with already existing scales.

Results – specific comment

Page 12

Table 5.  Authors need to explain why they adjusted the post test scores and the reason for including table 5.

Limitations and conclusion- specific comment

Page 16 Line 511

The authors do not acknowledge that all of the students who participated in the study volunteered to take the course – thus may have a better attitude toward aging to begin with.

As noted in the overall comments – Until more information is provided on the “control education” it is unclear what the results of the intervention are being compared to.

Round 2

Reviewer 2 Report

The authors have addressed my previous comments.  A good editorial review is still needed.  There are several points that need to addressed.  My comments were made to the pdf copy of the manuscript.  

Author Response

Attached please find the revised version. Thank you for your suggestion.
